# Would Older Adults Perform Preventive Practices in the Post-COVID-19 Era? A Community-Based Cross-Sectional Survey in China

**DOI:** 10.3390/ijerph181910169

**Published:** 2021-09-28

**Authors:** Meijun Chen, Xiaoqi Wang, Qingping Yun, Yuting Lin, Qingqing Wu, Qinghua Yang, Dezhi Wan, Dan Tian, Chun Chang

**Affiliations:** 1School of Public Health, Peking University, Beijing 100191, China; chenmeijun@pku.edu.cn (M.C.); yunqingping@bjmu.edu.cn (Q.Y.); 2011110174@bjmu.edu.cn (Y.L.); 2National Immunization Program, Chinese Center for Disease Control and Prevention, Beijing 100050, China; wangxq1@chinacdc.cn; 3Provincial Center for Disease Control and Prevention, Hangzhou 310006, China; qqwu@cdc.zj.cn; 4Provincial Health Education Center, Chongqing 401120, China; yaqh3@163.com; 5Provincial Patriotic Health and Health Promotion Center, Nanchang 330006, China; wdz739@163.com; 6Provincial Health Service Center, Shenyang 110005, China; sunliming.wsjkfwzx@ln.gov.cn

**Keywords:** older adults, post-COVID-19 era, preventive practices

## Abstract

During the post-COVID-19 era, preventive practices, such as washing hands and wearing a mask, remain key measures for controlling the spread of infection for older adults. This study investigated the status of preventive practices among older adults and identified the related influencing factors. Participants who were ≥60 years old were recruited nationwide. Data were collected through self-designed questionnaires, including demographic variables, knowledge, perceived vulnerability, response efficacy, anxiety and preventive practices. Descriptive statistics and chi-square tests were performed. Hierarchical logistic regression was conducted to determine the predictors. A total of 2996 participants completed this study. Of them, 2358 (78.7%) participants reported washing hands regularly in the last two weeks, and 1699 (56.7%) always wore masks outside this year. Knowledge (hand washing: OR = 1.09, *p* < 0.01; mask wearing: OR = 1.17, *p* < 0.01) and response efficacy (hand washing: OR = 1.61, *p* < 0.01; mask wearing: OR = 1.70, *p* < 0.01) were positively associated with preventive practices, whereas perceived vulnerability had a negative effect (hand washing: OR = 0.54, *p* < 0.01; mask wearing: OR = 0.72, *p* < 0.01). Knowledge, response efficacy and perceived vulnerability were found to be significant predictors of the preventive practice among older adults in the post-COVID-19 era. This study provides new insights into preventive suggestions after the peak of the pandemic and also has significant implications in improving the life quality of older adults.

## 1. Introduction

The outbreak of COVID-19 has posed a serious challenge to China and countries around the world. The Chinese government has adopted a series of strict measures to control the widespread of COVID-19, such as the lockdown of cities, compulsory mask wearing, quarantining people from high risks areas for 14 days, and promoting individual preventive behaviors [1]. On 8 May 2020, the State Council issued guidance on the Normalization of COVID-19 Epidemics Prevention and Control. Since then, the epidemic prevention and control across the country has entered a normal state and China has entered a post-COVID-19 era.

Personal hygiene measures, such as washing hands and wearing a mask, were proven to be effective in infection transmission reduction during the COVID-19 pandemic. Teslya et al. developed a deterministic compartmental transmission model to evaluate the impact of self-imposed measures (hand washing, mask wearing, and social distancing), which showed that self-imposed measures can prevent a large epidemic if their efficacy exceeds 50% [2]. A study in Hong Kong also found that preventive practices, including border restrictions, quarantine and isolation, distancing, and changes in population behavior, were associated with reduced transmission of COVID-19, with a 44% reduction in transmissibility in the community [3]. However, in the post-pandemic period, people were less likely to engage in the preventive behaviors that they performed during the pandemic. A Chinese research indicated that respondents reported a reduction in use and repeated uses of masks in the post-COVID-19 period [4], and a research study also reported that the proportion of people wearing masks dropped from 74.3% in March 2003 to 39.1%, when SARS was ending three months after in Hong Kong [5].

During the post-COVID-19 period, although the large-scale epidemic may have ended in China, the emergence of sporadic new cases continues to pose a risk for future outbreaks. Older patients with COVID-19 have higher rates of severe disease and death than younger patients [6], and they are usually evaluated as a high-risk group due to their poor health status and low immune capacity. A Japanese study showed that 47.3% of older adults became less active during the COVID-19 epidemic, resulting in a lower health-related quality of life and mental health [7]. Vaccines are believed to be one of the most effective ways to contain the pandemic and prevent new outbreaks [8] in the post-pandemic period. Previous data showed a reduction in COVID-19 cases and severe illness in populations with high vaccination coverage [9]. A research study from the CDC of United States found that receiving the COVID-19 vaccine was 64% effective against COVID-19 hospitalization among partially vaccinated adults aged ≥ 65 years and 94% effective among fully vaccinated adults aged ≥ 65 years [10]. Since 14 March 2021, some Chinese regions have started vaccinating people over 60 who are in good health. However, in the older Chinese population, COVID-19 vaccination coverage rates remain relatively low. Many researchers found that the immunogenicity of the vaccine decreased with age, eliciting lower overall humoral responses in adults 65 to 85 years of age compared to those between 18 and 55 years of age [11,12,13]. Thus, the preventive practices toward COVID-19, such as hand washing and mask wearing, remain key measures for controlling the spread of infection for older adults.

Many studies have explored the relationship between knowledge, attitudes and prevention measures during the epidemic period, but few studies have focused on the protective behaviors of older adults in the post-epidemic era. Based on protection motivation theory (PMT), when individuals encounter a threatening event, they are primarily motivated to engage in protective behavior [14]. During global pandemics, such as the COVID-19 pandemic, people experience fear and anxiety and realize that there is no definitive treatment for the disease. This kind of fear is appraised to predict and encourage protective behaviors [15,16]. Therefore, this study aimed to investigate the status of the knowledge, beliefs and preventive practices toward COVID-19 among older adults in the post-COVID-19 period, and identify factors influencing preventive practices, in order to provide preventive suggestions for combating the spread of COVID-19 and improve the life quality of older adults.

## 2. Materials and Methods

### 2.1. Study Design and Participants

This was a cross-sectional study conducted by The School of Public Health, Peking University (PKU) from May to September in 2020, after the lockdown period of the COVID-19 pandemic, when measures of regular epidemic prevention and control were taken by the government. A multi-stage, cluster sampling method was used. First, two provinces were randomly selected from the east, west and middle parts of mainland China. Second, the provincial capital cities and other random cities (below the median GDP) were selected from each province; overall, a total of 3056 participants from Zhejiang, Jiangxi, Chongqing, Gansu, Liaoning, and Beijing were selected to participate in this study. The inclusion criteria of the participants were as follows: (1) ≥60 years old; and (2) agreed to participate in the survey. Participants with dementia or mental disorders were excluded.

Participants were recruited from community health care stations with the help of family doctors via social media (i.e., WeChat group, and public service announcements) and community outreach (i.e., distribution of flyers, personnel visits to health care settings and participation in health education activities). Some participants were also recruited by telephone based on the patients’ healthcare record.

### 2.2. Data Collection

Before the interview, investigators introduced the purpose, content, benefits and risks of participating in the research to the participants, and then the informed consent was signed by the participant. Participants who were not able to read and write were informed of the background, purpose, steps, risks and benefits of this study by a face-to-face interview, and they had enough time and opportunities to ask questions. Although they were unable to read and write, most of them had the ability to write their names. According to *Ethical considerations in biomedical research involving human beings* published by National Health Commission of the People’ Republic of China in 2016, If people cannot provide the written consent, investigators should obtain oral informed consent and submit supporting materials.

In order to implement quality control, the investigators received unified training before the interview so that they had a full understanding of our study. All investigators received a manual. When they encountered any problems during the investigation, they were able to consult this manual. In principle, the respondents should answer according to their own understanding and complete the questionnaire on their own. It was strictly forbidden to explain the answer or answer the question for the respondents. If the respondent has difficulties in reading or writing, and could not complete the questionnaire independently, the investigator would ask the question by a face-to-face inquiry. The investigator completed the questionnaire according to the response of the participants. The investigators were not allowed to use inductive or suggestive language. If the respondent had a low level of education, investigators were permitted to provide an appropriate explanation, but the explanation had to be faithful to the original intent.

The questionnaire was designed by the research team in PKU. It comprised the following main sections: socio-demographic variables, knowledge, perceived vulnerability, response efficacy, anxiety and preventive practices toward COVID-19. the socio-demographic variables collected included age, gender, education, marital status, income, region, household composition and medical insurance. The knowledge questions covering issues such as isolation period and precautions in COVID-19 ranged from 0 to 8. Perceived vulnerability and response efficacy were PMT constructs. Perceived vulnerability was assessed by asking if the participants were aware of the high risk of being infected in the future, using a 4-point scale (1 = impossible, 4 = very possible); the items also included a “Don’t know” option. Response efficacy was assessed by the following question: do you believe that personal preventive practices are effective? Anxiety was also an important factor based upon studies testing the psychological impact of preventive behaviors during SARS and COVID-19 [17,18]. To assess anxiety, participants indicated on a 1 (“not at all”) to 3 (“Very”) scale whether they felt anxiety or panic since the outbreak of the epidemic. Preventive behaviors to mitigate the spread of COVID-19 during the post-COVID-19 period were measured by 2 variables: how often did you wash hands in the past 2 weeks, and how often do you wear a mask outdoors? Response options ranged from 1 (“Never”) to 5 (“Always”).

The data were collected through the questionnaires at community health care stations by trained and experienced enumerators. The survey took approximately 15–20 min for each participant to complete. A total of 3056 questionnaires were returned. For paper questionnaires, they were considered invalid if there were a large number of unfinished questions. If they completed the questionnaire on a tablet, answers with a completion time of less than 5 min, shown by back-end data, were considered low quality and invalid. After excluding those illegible questionnaires, the final study sample was 2996. Confidentiality was maintained by using study codes. Data access was restricted to study researchers only.

### 2.3. Data Analysis

Descriptive statistics were calculated for all participant characteristics. Categorical variables are presented as the frequency (*n*) and percentage (%). Associations between knowledge, perceived vulnerability, response efficacy, anxiety toward COVID-19 and demographic characteristics were examined in chi-square tests. Hierarchical logistic regression was conducted to examine factors associated with preventive practices toward COVID-19, such as knowledge, beliefs and demographic characteristics. Multivariate logistic regression was performed to determine the predictors of preventive practices toward COVID-19. Statistical significance was indicated by *p* < 0.05. Data management and analysis were performed using SPSS 24.0.

## 3. Results

### 3.1. Socio-Demographic Characteristics of the Study Participants

The socio-demographic characteristics of study participants were shown in Table 1. A total of 2996 older adults completed this study. The participants’ mean age was 69.3 ± 6.5 years. A total of 1672 (55.8%) were female. Almost one-third completed primary school (32.2%), and 890 (29.7%) completed middle school. Most of the participants were married (81.7%). The proportion of respondents who lived in an urban area was 51.6%. A higher proportion of respondents (46.7%) reported a low- and middle income.

### 3.2. Knowledge, Perceived Vulnerability, Response Efficacy and Anxiety towards COVID-19 among Old Adults

Table 2 presents the details of the knowledge and beliefs toward COVID-19. Participants with a score greater than four were considered to have a good knowledge of COVID-19 preventive measures. The proportion of urban older adults who had a good knowledge of COVID-19 preventive measures was significantly higher than that of rural older adults (*χ*^2^ = 27.233, *p* < 0.01). Participants with a higher education level and household income were observed to have a better score of COVID-19 knowledge (*χ*^2^ = 47.90, *p* < 0.01; *χ*^2^ = 41.16, *p* < 0.01).

A total of 815 female (48.7%) participants felt anxiety or panic since the outbreak of the epidemic (*χ*^2^ = 6.134, *p* = 0.01). People who lived in rural area (49.9%) and with a low income (52.1%) reported a higher proportion of being anxious (*χ*^2^ = 11.09, *p* < 0.01; *χ*^2^ = 15.09, *p* < 0.01). There was no difference in the reports of perceived vulnerability. In terms of response efficacy, 1329 (85.9%) urban residents believed that personal preventive practices were effective (*χ*^2^ = 23.31, *p* < 0.01). Participants with higher education level and household income reported a better response efficacy of preventive practices (*χ*^2^ = 88.67, *p* < 0.01; *χ*^2^ = 46.63, *p* < 0.01). People with one or more chronic diseases reported better knowledge, that they felt anxiety and that they believed that personal preventive practices were effective.

### 3.3. Preventive Practices toward COVID-19 among Older Adults

A total of 2358 (78.7%) participants reported having washed their hands regularly in the last two weeks, and 1699 (56.7%) always wore masks outside this year. Figure 1 shows the percentage of participants who had washed their hands regularly in the last two weeks. Urban residents reported a higher rate of hand washing in the last two weeks (90.2%) compared with rural residents (66.4%), with a statistical difference (*p* < 0.01). Figure 2 shows the percentage of participants who always wore masks outside. The rate of urban older adults (68.8%) was significantly higher than that of rural older adults (43.8%, *p* < 0.01). Participants with a higher education level and household income reported better performance in hand washing and mask wearing (*p* < 0.01).

### 3.4. Factors Associated with Preventive Practice toward COVID-19

Table 3 shows the results of a hierarchical logistic regression analysis of factors related to hand washing. In model 3, the education level, income level, and region are significantly associated with hand washing (*p* < 0.05). Participants who were younger, lived in an urban area, received high school or higher education, and had higher income were more likely to have washed their hands regularly in the last two weeks. Knowledge (Model 2: OR = 1.11, *p* < 0.01; Model 3: OR = 1.09, *p* < 0.01) and response efficacy (OR = 1.61, *p* < 0.01) were positively associated with the practice of hand washing. Perceived vulnerability was negatively associated with the practice of hand washing (OR = 0.54, *p* < 0.01).

Table 4 shows the results of a hierarchical logistic regression analysis of factors related to mask-wearing. In all models, a positive and significant association was observed between education level, income level, the number of chronic diseases and the practice of wearing masks outside. Urban older adults were more likely to wear masks when they went outside. Knowledge (Model 2: OR = 1.19, *p* < 0.01; Model 3: OR = 1.17, *p* < 0.01) and response efficacy (OR = 1.70, *p* < 0.01) were positively associated with the practice of mask wearing. Nevertheless, perceived vulnerability was negatively associated with the practice of mask wearing in the final model (OR = 0.72, *p* < 0.01). Knowledge regarding COVID-19, response efficacy and perceived vulnerability were found to be significant predictors of the preventive practices among older adults during the post-COVID-19 period.

## 4. Discussion

This study focused on the adoption of preventive behaviors among the older Chinese population. This paper extended the prior research along two dimensions. First, although many studies have explored the relationship between knowledge, attitudes and prevention measures during the epidemic period, few have focused on the protective behaviors in the post-epidemic period. Thus, to our best knowledge, this is the first study that explored the factors influencing preventive practices after the peak of the pandemic in the older Chinese population. Second, PMT was applied to our study. Knowledge and response efficacy were found to be positive predictors, whereas perceived vulnerability was observed to be a negative predictor. This suggests that a successful and sustainable strategy for combating the spread of COVID-19 should not only provide information, but might also need to lower the risk estimates and focus on the mental health, which provides new insights into preventive suggestions for combating the spread of COVID-19 after the peak of the pandemic.

In the post-COVID-19 period, the percentage of respondents frequently wearing masks declined sharply, compared with that during the early outbreak period. In a Chinese research study conducted from January to February in 2020 (early outbreak of the pandemic), 97.9% participants used face masks in public [19]. Another study also revealed the high frequency of wearing a face mask in public venues (*n* = 1035, 81.9%) among Chinese older adults [6]. However, in our study, only 56.7% of older adults reported that they always wore masks when going outside this year; this finding is consistent with a similar study regarding SARS [5]. As for the predictors of mask wearing and knowledge regarding COVID-19, response efficacy and perceived vulnerability were found to be significant predictors among older adults in our study. Good precautionary measures were also detected among medical students who have good levels of knowledge regarding COVID-19 as well as positive attitudes toward the disease [20]. Beaudoin et al. found that preventive behavior was positively associated with severity, self-efficacy, and response efficacy [21]. Coroiu et al. suggested that individuals are motivated to engage in preventive practices by civic responsibility, including wanting to protect themselves and others, wanting to avoid spreading the virus to others and feeling a responsibility to protect the community [22]. Wearing a face mask is not only a personal preventive choice, but is also a prevention measure mandated by the government. Gotanda et al., found that individuals with high trust in government practiced preventive measures against COVID-19 more often than those with low trust at the national level [23]. Individuals are more likely to adopt restrictive preventive measures when they trust the government and believe that it is working for the best interests of the population to curb the spread of COVID-19. Nevertheless, during the post-COVID-19 period, wearing a mask is no longer a mandatory measure in many places, but it is still necessary for older adults due to potential recurrent epidemics. Inner factors, rather than the mandatory prevention measure by the government, seem to have a greater impact on whether people wear masks in the post-COVID-19 period, which may be key factors for improving preventive behavior.

It was found that 78.7% participants reported good hand hygiene in the last two weeks, which was maintained at a relatively low level compared with previous pandemics in China. For example, the percentage of subjects with frequent/very frequent hand-washing behavior was 93.9% 3 months after the end of the SARS epidemic in Hong Kong in 2003 [5]. A study conducted to monitor the hand-washing behavior after the outbreak of the H5N1 influenza virus showed 86.7% respondents with regular hand-washing behaviors in 2005 [24]. However, compared with other countries, the percentage of subjects with frequent/very frequent hand washing in China was higher. In Spain, only 16.7% people reported that they washed their hands more frequently than before the influenza A/H1N1 pandemic one year after the pandemic [25]. Nevertheless, the level of frequency of hand washing found in this study was less than other previously conducted studies during the pandemic. Huang et al. found that 97.9% of respondents believed that they wash their hands more often than usual [19]. These findings are consistent with those observed in previous studies on the SARS and H5N1 influenza outbreaks [24,26]. Unlike wearing a face mask, hand washing is more likely to be a personal health-seeking behavior. Knowledge, response efficacy and perceived vulnerability were significant predictors of the practice of hand washing. Therefore, developing good personal hygienic awareness among older adults and motivating them to maintain personal health-seeking behavior are urgent challenges in the post-COVID-19 period.

According to *PMT*, fear is appraised to encourage protective behaviors when people are faced with threats. If the perceived severity and vulnerability are high, and the perceived rewards are low, there is a stronger motivation for engagement in health-promoting behaviors [15]. In the previous studies about preventive measures of COVID-19, Yazdanpanah et al. [27] and Hromatko et al. [28] reported that perceived vulnerability produced a positive and significant impression on preventive behaviors. However, in our study, a negative association was observed between perceived vulnerability and preventive behaviors. It could be explained by adaptive accuracy of risk perceptions. Brewer et al. noted that increases in preventive behaviors from T1 (initial point in time) to T2 (subsequent point in time) were significantly associated with a decrease in perceived risk in a longitudinal research of an acute livestock epidemic [29]. Participants who have taken precautions to reduce their risk will perceive themselves as being less at risk, indicating adaptive decreases in risk perceptions [30]. In other words, perceptions of risk encourage people to take preventive measures in order to reduce this risk, which shows a negative relation between perceived vulnerability and preventive behaviors in the post-pandemic period. Thus, during the post-COVID-19 period, people who believed that wearing masks and washing hands could effectively reduce individual risk and had taken actions would lower their personal risk perceptions. This kind of negative association is in agreement with the results reported by Klepper et al. [31] and Chen et al. [32] in alcohol- and cigarettes-related preventive behaviors.

Our study also has several limitations. First, preventive practices are particularly relevant with the prevention and control policy taken by the local government. Wearing masks in public places is mandatory in some cities, whereas in some cities, it is not. Policy factors have not been taken into consideration in this study. Second, the patients recruited in this study were voluntarily enrolled or recruited from family doctors; thus, there may be a volunteer bias, affecting the representativeness of the research. However, as the participants were recruited from different provinces of China, it is believed that the overall findings are meaningful. Third, we only used cross-sectional data for this estimation. As a result, a causal relationship could not be inferred. Moreover, there are many other face-mask-related issues that we have not discussed in this article, such as the reusing of masks [4] and face-mask-related adverse skin reactions [33]. In the future, it will be necessary to expand the scope of these related issues.

Despite these limitations, this study has some important implications. The results indicated the status of the knowledge, beliefs, preventive practices toward COVID-19 among older adults, which demonstrates the need for health behavior maintenance in the post-COVID-19 period. Our results imply that knowledge toward COVID-19 and response efficacy are positive predictors in older adults. Accordingly, it is important for health service providers to provide popular, persistent, and trustworthy knowledge targeted at older adults. In addition, older adults who perceived a low vulnerability reported better performance in mask wearing and hand washing. If people adopted preventive practices and decreased their risk perceptions as a consequence, they would be more motivated to maintain the behavior pattern. From a health education perspective, these results suggest that successful and sustainable strategy for combating the spread of COVID-19 should not only provide information, but might also need to lower the risk estimates and focus on mental health.

## 5. Conclusions

In summary, participants who lived in an urban area, and had a higher education level and household income were observed to have a better score of COVID-19 knowledge and response efficacy of preventive practices. Females, rural residents and participants with a low income reported a higher proportion of being anxious. Moreover, urban residents and participants with a higher education level and household income reported better performance in hand washing and mask wearing. In addition, knowledge regarding COVID-19 and response efficacy were found to be significant and positive predictors, and perceived vulnerability was negatively associated with preventive practice among older adults during the post-COVID-19 period. This study provides new insights into preventive suggestions for combating the spread of COVID-19 after the peak of pandemic and also has significant implications in improving the life quality of older adults.

## Figures and Tables

**Figure 1 ijerph-18-10169-f001:**
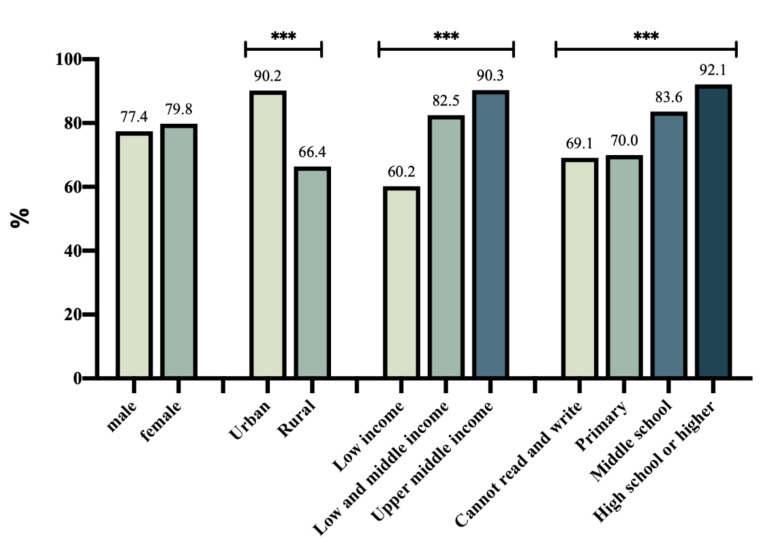
Percentage of participants who had washed hands regularly in the last 2 weeks. (***, *p* < 0.01).

**Figure 2 ijerph-18-10169-f002:**
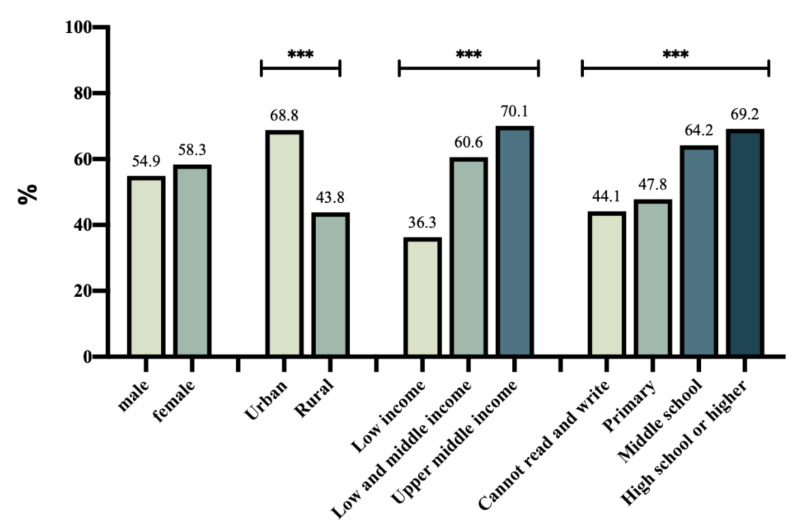
Percentage of participants who always wore masks outdoors. (***, *p* < 0.01).

**Table 1 ijerph-18-10169-t001:** The socio-demographic characteristics of the study participants.

Characteristics	Participants
Age	
60~69	1737 (58.0)
≥70	1259 (42.0)
Gender ^a^	
male	1317 (44.0)
female	1672 (55.8)
Education	
Illiterate/Barely literate	493 (16.4)
Primary	964 (32.2)
Middle school	890 (29.7)
High school or higher	649 (21.7)
Marital status ^b^	
married	2439 (81.7)
Unmarried/divorce/widowed	547 (18.0)
Household income ^c^	
Low	791 (26.4)
Low and middle	1398 (46.7)
Upper middle	796 (26.7)
Region ^d^	
Urban	1547 (51.6)
Rural	1448 (48.3)
Household composition	
Living with others	2736 (91.3)
Living alone	260 (8.7)
Medical insurance	
No insurance	40 (1.3)
Basic old-age insurance for urban workers	999 (33.3)
Basic medical insurance for urban residents	837 (27.9)
Rural cooperative medical care	1062 (35.4)
Others (commercial insurance, state medicine, etc.)	161 (5.4)
Number of chronic diseases	
0	1212 (40.5)
1	1014 (33.8)
≥2	761 (25.7)
Total	2996

^a^ Missing data for 7 participants. ^b^ Missing data for 10 participants. ^c^ Income level was classified as low, low and middle, average and upper middle. For urban residents (personal monthly income): low, RMB < 600 (USD 91); low and middle, RMB 600~3500 (USD 91~533), and upper middle, RMB ≥ 3500 (USD 533). For rural residents (household annually income): low, RMB < 17,000 (USD 2589); low and middle, RMB 17,000~65,000 (USD 2589~9900) and upper middle, RMB ≥ 65,000 (USD 9900). ^d^ Missing data for 11 participants. Missing data for 1 participant.

**Table 2 ijerph-18-10169-t002:** Knowledge, anxiety, perceived vulnerability and response efficacy toward COVID-19 among older adults in different groups (n,%).

	Knowledge Score			Felt Anxiety or Panic Since the Outbreak of Epidemic			Perceived High Risk of Being Infected in the Future			Believed That Personal Preventive Practices Were Effective		
	≥4	*χ* ^2^	*p*	Yes	*χ* ^2^	*p*	Yes	*χ* ^2^	*p*	Yes	*χ* ^2^	*p*
Gender		0.258	0.612		6.134	0.013		0.184	0.668		0.015	0.904
Male	1127, 85.6			582, 44.2			104, 7.9			1090, 82.8		
Female	1441, 86.2			815, 48.7			125, 7.5			1381, 82.6		
Region		27.233	0.000		11.092	0.001		1.771	0.183		23.305	0.000
Urban	1379, 89.1			678, 43.8			128, 8.3			1329, 85.9		
Rural	1194, 82.5			722, 49.9			101, 7			1148, 79.3		
Education level		47.896	0.000		1.756	0.625		3.973	0.264		88.674	0.000
Illiterate	379, 77			218, 44.3			46, 9.3			353, 71.7		
Primary	827, 85.8			450, 46.7			63, 6.5			766, 79.5		
Middle school	776, 87.2			426, 47.9			72, 8.1			767, 86.2		
High school or higher	591, 91.1			306, 47.1			48, 7.4			591, 91.1		
Income level		41.156	0.000		15.089	0.001		0.317	0.853		46.627	0.000
Low	628, 79.4			412, 52.1			60, 7.6			594, 75.1		
Low and middle	1230, 88			646, 46.2			111, 7.9			1186, 84.8		
Upper middle	710, 89.2			338, 42.5			58, 7.3			692, 86.9		
Number of chronic diseases		15.200	0.001		7.642	0.022		0.068	0.967		20.330	0.000
0	1011, 83.4			531, 43.8			94, 7.7			971, 80.1		
1	871, 85.9			500, 49.3			76, 7.5			828, 81.7		
≥2	684, 89.9			368, 48.4			58, 7.6			671, 88.2		

**Table 3 ijerph-18-10169-t003:** Hierarchical logistic regression results on factors related to hand washing.

Predictors	Model 1				Model 2				Model 3			
	B	Wald	Exp(B)	*p*-Value	B	Wald	Exp(B)	*p*-Value	B	Wald	Exp(B)	*p*-Value
Gender	0.198	3.773	1.219	0.052	0.184	3.230	1.202	0.072	0.172	2.777	1.187	0.096
Age	−0.012	2.350	0.988	0.125	−0.012	2.223	0.988	0.136	−0.012	2.149	0.989	0.143
Education		22.949		0.000		20.560		0.000		18.158		0.000
Primary	−0.050	0.147	0.951	0.701	−0.128	0.936	0.880	0.333	−0.167	1.550	0.847	0.213
Middle school	0.281	3.525	1.324	0.060	0.188	1.530	1.206	0.216	0.132	0.740	1.141	0.390
High school or higher	0.772	14.808	2.164	0.000	0.664	10.755	1.943	0.001	0.582	8.115	1.790	0.004
Marital status	−0.126	1.007	0.882	0.316	−0.076	0.358	0.927	0.550	−0.104	0.659	0.902	0.417
Income level		40.416		0.000		37.851		0.000		33.757		0.000
Low and middle	0.659	32.939	1.933	0.000	0.638	30.524	1.892	0.000	0.601	26.600	1.824	0.000
Upper middle	0.897	27.196	2.451	0.000	0.887	25.888	2.404	0.000	0.846	23.845	2.330	0.000
Region	−0.874	44.917	0.417	0.000	−0.875	45.092	0.417	0.000	−0.924	49.502	0.397	0.000
Chronic disease	−0.038	0.047	0.963	0.428	−0.056	1.385	0.945	0.239	−0.071	2.136	0.932	0.144
COVID-19 knowledge					0.108	18.795	1.114	0.000	0.084	10.578	1.088	0.001
Response efficacy									0.474	15.033	1.607	0.000
Anxiety										0.386		0.824
A little									0.099	0.386	1.104	0.534
Very									0.019	0.001	1.020	0.978
Perceived vulnerability									−0.614	13.447	0.541	0.000
Model Chi-square	344.992				363.495				390.456			
Degree of freedom	10				11				15			
Model significance	<0.001				<0.001				<0.001			

**Table 4 ijerph-18-10169-t004:** Hierarchical logistic regression results on factors related to mask wearing.

Predictors	Model 1				Model 2				Model 3			
	B	Wald	Exp(B)	*p*-Value	B	Wald	Exp(B)	*p*-Value	B	Wald	Exp(B)	*p*-Value
Gender	0.180	4.725	1.198	0.030	0.163	3.795	1.177	0.051	0.158	3.496	1.171	0.062
Age	−0.007	1.136	0.993	0.287	−0.006	0.816	0.994	0.366	−0.006	0.808	0.994	0.369
Education		18.108		0.000		11.177		0.011		8.652		0.034
Primary	0.116	0.951	1.123	0.329	0.009	0.006	1.009	0.939	−0.027	0.050	0.973	0.824
Middle school	0.455	12.702	1.577	0.000	0.321	6.063	1.379	0.014	0.266	4.069	1.305	0.044
High school or higher	0.443	9.103	1.558	0.003	0.281	3.500	1.324	0.061	0.196	1.672	1.217	0.196
Marital status	−0.012	0.014	0.988	0.906	0.051	0.222	1.052	0.637	0.040	0.136	1.041	0.713
Income level		47.336		0.000		43.283		0.000		39.620		0.000
Low and middle	0.655	38.443	1.924	0.000	0.633	35.186	1.884	0.000	0.605	31.545	1.831	0.000
Upper middle	0.856	39.633	2.354	0.000	0.828	36.224	2.289	0.000	0.807	33.819	2.240	0.000
Region	−0.460	20.643	0.631	0.000	−0.466	20.722	0.628	0.000	−0.492	22.696	0.611	0.000
Chronic disease	0.134	12.208	1.143	0.000	0.108	7.701	1.114	0.006	0.106	7.171	1.112	0.007
COVID-19 knowledge					0.177	62.104	1.193	0.000	0.156	45.857	1.168	0.000
Response efficacy									0.530	23.182	1.699	0.000
Anxiety										4.731		0.094
A little									−0.213	2.686	0.808	0.101
Very									0.918	1.869	2.505	0.172
Perceived vulnerability									−0.336	5.138	0.715	0.023
Model Chi-square	287.245				351.328				383.658			
Degree of freedom	10				11				15			
Model significance	<0.001				<0.001				<0.001			

## Data Availability

The data presented in this study are available on request from the corresponding author.

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
