# Peer review of "Would Older Adults Perform Preventive Practices in the Post-COVID-19 Era? A Community-Based Cross-Sectional Survey in China"

_ijerph, 2021, doi:10.3390/ijerph181910169_

Round 1

Reviewer 1 Report

We congratulate the authors for writing the paper and carrying out the study.

The paper has importance for the researchers/readers; complies with the scientific structure, is well-referenced, with current references and are adjusted to the theme.

In the introduction, the authors adequately present the problem. Ideas have a logical presentation and are well framed.

The methodology is clearly explained There are sufficient details to replicate the proposed experimental procedures and analysis.

The results are well organized using tables and graphs.
The discussion, like the conclusion, is pertinent, clear and focuses on fundamentals points.

Here are some small details to be analyzed by the authors:
1)    Line 4 to 13 - to standardize the use of the letters (a, b, ...) with the affiliations that are presented in numbers.
2)  we suggest putting the information in line 26/27 as follows: hands-washing: OR = 0.54, p < 0.01 mask-wearing: OR = 0.72, p < 0.0 5).

3)  line 44 - delete the first name of the cited author. Teslya et al instead of Alexandra Teslya. Review the other situations throughout the paper.

4) to standardize the concepts. In the title and abstract the authors use old adults, but later in the text they tend to use elderly. elderly is a concept less and less accepted, so our suggestion is to adopt the use of the concept  "old adults".5)    The assessment of the study by the ethics committee is not expressed in the paper. it's important to do it!
6)    In the legend of table 1, it is necessary to place the letters, such as: a - Missing data 7….
7)    Line 189 -  we would like the authors to confirm whether the phrase - "In model 3, age, educational level ..." corresponds to the data presented in table 3,  with regard to age.
8)    Line 216 - since the authors presented the acronym PTM on line 78, it now makes sense to use it, like line 269
9)    the authors say in line 257 that "In Spain, there was only 16.7% people (…)  pandemic [25]....", but reference 25 refers to a French study, right?

best regards 

Author Response

Thanks  for all your kind and nice suggestion.

Reviewer 2 Report

Dear authors,

Please provide answers to my questions to allow a fair, complete review of your manuscript. 

Page 2 line 96- How the selection process did unfold?

Page 3-Data collection

  • Information on page 3 lines 100-103 must be moved to recruitment in the previous section.
  • How did you contact the prospective participants?
  • Which are the reasons for questionnaires illegible?
  • How free, voluntary participation was ensured?
  • How the power influence from health professionals over the prospective participants were controlled?
  • What was the refusal rate of participation?
  • What was the education profile of seniors older than 60 years in these research sites?
  • Were they all literate and then skilled to provide a signed informed consent?

Page 4

  • 4% of the participants were unable to read and write so how they were able to provide a signed informed consent?
  • How the data collection was conducted with them if they were unable to read the questionnaires?
  • What were the overall health conditions of the participants? Chronic conditions is briefly introduced in page 7
  • Any awareness of being of risk person to Covid?
  • Would health conditions a factor for perceived vulnerability, anxiety and motivation to self-protection?

Author Response

Thank you so much for your helpful comments.

Round 2

Reviewer 2 Report

Many thanks for the additional information that improved the overall quality of your manuscript. I just want to suggest you to add some reference/comment about the ethics procedure of getting digital imprint as a form of authorization in a no-read document by the individual who provides the authorization. Which Chinese document does support the use of this procedure? There is a unnecessary sentence: Once found, the relevant questionnaire would be deleted. It may imply a major gap in the training for data collection.